# Impact of Surgical Intervention on Nonobstructive Dysphagia: A Retrospective Study Based on High-Resolution Impedance Manometry in a Taiwanese Population at a Single Institution

**DOI:** 10.3390/jpm12040590

**Published:** 2022-04-07

**Authors:** Gang-Hua Lin, Kuan-Hsun Lin, Szu-Yu Lin, Tsai-Wang Huang, Hung Chang, Hsu-Kai Huang

**Affiliations:** 1Department of Surgery, Tri-Service General Hospital, National Defense Medical Center, Taipei 114, Taiwan; a2015joy@gmail.com; 2Division of Thoracic Surgery, Department of Surgery, Tri-Service General Hospital, National Defense Medical Center, Taipei 114, Taiwan; chi-wang@yahoo.com.tw (T.-W.H.); hung@ndmctsgh.edu.tw (H.C.); 3Critical Care Medicine, Tri-Service General Hospital, National Defense Medical Center, Taipei 114, Taiwan; missyu0812@gmail.com

**Keywords:** high-resolution impedance manometry, achalasia, esophagogastric junction outflow obstruction, Chicago Classification, Eckardt score

## Abstract

Esophageal motility disorders account for a large proportion of nonobstructive dysphagia cases, which constitute a heterogeneous group of diagnoses that commonly result in peristaltic derangement and impaired relaxation of the lower esophageal sphincter. We performed a single-institution retrospective study enrolling consecutive patients with chief complaints of dysphagia who underwent HRIM from December 2014 to December 2019, and analyzed demographic, clinical, and manometric data using descriptive statistics. In total, 277 identified patients were included in the final analysis. Ineffective esophageal motility (*n* = 152, 24.5%) was the most common diagnosis by HRIM, followed by absent contractility, EGJ outflow obstruction, type II achalasia, and type I achalasia. Furthermore, surgery including exploratory, laparoscopic, and robotic myotomy, as well as POEM, is considered the most effective treatment for patients with non-spastic achalasia and EGJOO, due to its effective symptom palliation and prevention of disease progression; surgery also contributes to an obvious improvement of dysphagia compared with slightly less efficacy for other related symptoms. Our study aimed to elaborate the clinical characteristics of patients with nonobstructive dysphagia based on HRIM in a Taiwanese population, and to analyze the therapeutic outcomes of such patients who ultimately underwent surgical interventions.

## 1. Introduction

Dysphagia is a subjective sensation of difficulty or abnormality with swallowing, which can be generally divided into categories including esophageal and oropharyngeal etiologies. Esophageal dysphagia can be separated into mechanical obstruction, which initially affects only solids, and localized neuromuscular motility disorders, in which both solids and liquids can be blocked. As the main causes of nonobstructive dysphagia, esophageal motility disorders include a heterogeneous group of diagnoses that commonly meet a common end result of peristaltic derangement and impaired relaxation of the lower esophageal sphincter (LES). Clinical manifestations of esophageal motility disorders are varied, but dysphagia, regurgitation, chest pain, and related body weight loss are the most commonly reported symptoms [1].

High-resolution impedance manometry (HRIM) assesses pressure changes in the esophageal body and LES, and is now considered the gold standard for the diagnosis of esophageal motility disorders. It provides improved and more detailed information compared to conventional manometry—particularly esophageal pressure topography, which enables quick and intuitive interpretation [2,3]. Chicago Classification version 3.0 is an updated algorithmic scheme utilization for analyzing clinical HRIM studies, which has since been applied to interpret manometric findings and facilitate a diagnosis of esophageal motility disorders [4]. Achalasia is a disorder characterized by the absence of peristalsis and defective relaxation of the LES, resulting in impaired bolus transport and stasis of food in the esophagus. Subclassification of achalasia includes type I, which is characterized by the absence of pressure waves recorded in the distal esophagus; type II, featuring panesophageal pressurizations; and type III, with at least 20% of swallows revealing rapidly propagating or spastic simultaneous contractions. EGJ outflow obstruction (EGJOO), which is similar to achalasia in exhibiting incomplete LES relaxation, involves intact or weak peristalsis [2].

Conceptualizing esophageal motility disorders as primarily non-spastic—which includes type I and II achalasia and EGJOO—or spastic—which consists of type III achalasia, hypercontractile (jackhammer) esophagus, and distal esophageal spasm (DES)—can help inform the choice of initial and subsequent treatments [5]. Surgical intervention is considered the most effective treatment for patients with non-spastic achalasia and EGJOO, characterized by splitting the fibers of the esophageal muscle layer through a longitudinal cut to reduce the pathologically elevated pressure at the LES, thus correcting insufficient relaxation and improving quality of life [6]. In contrast, patients with spastic motility disorders should undergo an initial trial of smooth muscle relaxants (typically with calcium channel blockers or nitrates) before being offered invasive interventions [7,8].

The Eckardt score is the most commonly used metric for evaluating achalasia therapies, even though it plays no role in differentiating diagnosis. Gockel et al. demonstrated that the Eckardt score tends toward being the most useful system for clinical practice because of its wide range and interval-level measurement properties converting the score to the Eckardt stages [9]. Furthermore, Shemmeri et al. concluded that the use of the Eckardt score to assess outcomes revealed outstanding results after the surgery [10].

At present, a few studies have clarified the etiologies of nonobstructive dysphagia patients in a Taiwanese population [11], but no further therapeutic conclusion has been demonstrated. Here, we present the clinical characteristics of patients with nonobstructive dysphagia based on HRIM in a Taiwanese population analyzed in our study. The main aim of our research was to elaborate the therapeutic outcomes of those patients who ultimately underwent surgical interventions.

## 2. Materials and Methods

### 2.1. Study Design and Patients

We performed a single-institution retrospective study enrolling consecutive patients with chief complaints of dysphagia undergoing HRIM from December 2014 to December 2019 at the Tri-Service General Hospital (TSGH). We excluded patients with oropharyngeal dysphagia, secondary dysphagia, or prior upper gastrointestinal surgical interventions. The patients involved were all given a questionnaire to determine their Eckardt scores before HRIM. Esophageal motility disorders were interpreted using the Chicago Classification version 3.0 [4]. This study was approved by the Institutional Review Board of the TSGH in 2021 (TSGHIRB No.: A202005190).

### 2.2. Symptom Evaluation

The Eckardt score is the most commonly used metric for evaluating achalasia therapies. Questions were asked about the frequency of esophageal symptoms based on the patient’s self-reported response, including dysphagia, regurgitation, and retrosternal pressure sensations. Depending on whether any of these symptoms occurred never, occasionally, daily, or with each meal, a symptom score between 0 and 3 was applied. In addition, a symptom score of 0–3 was assigned to the degree of body weight loss, corresponding to none, <5 kg, 5–10 kg, and >10 kg, respectively [12]. Therefore, the total score could vary from 0 to 12. The following clinical stages were defined: stage 0, score of 0–1; stage I, score of 2–3; stage II, score of 4–6; and stage III, score of >6. Clinical success was defined as a score of 3 or lower after the therapeutic intervention [13].

### 2.3. Esophagoscopy or Upper Gastrointestinal Series

These evaluations are crucial imaging tools used to exclude mechanical and inflammatory lesions that present relevant symptoms. Esophagoscopy directly reveals food retention inside the esophageal lumen, and evaluates distal esophageal and EGJ resistance until advancing the endoscope into the stomach, further excluding obstructive lesions.

Upper gastrointestinal series were performed first via the oral administration of high-density barium sulfate diluted commercially with 70 mL of water, and the patients were asked to turn 360° to coat the gastric fundus. Multiple views were taken, including upright left posterior oblique followed by mucosal relief views, recumbent right lateral views, prone right anterior oblique views, and dynamic views of the supine to right posterior oblique and right lateral positions under fluoroscopy [14]. Valuable information regarding the functional evaluation of the pharynx, mucosal abnormalities of the esophagus, and motility disorders was interpreted.

### 2.4. Esophageal HRIM

After an overnight fast, patients underwent esophageal HRIM performed by an experienced technician. The HRIM probe was 12 French in diameter, with 32 closely spaced water-perfused pressure sensors and 16 impedance channels (InSIGHT Ultima^®^ High-Resolution Manometry, Diversatek Healthcare, MedDataLink Gastroenterology (MDLGI), Milwaukee, Brookfield, WI, USA). The catheter was introduced transnasally after being zeroed to atmospheric pressure, and the patients were asked to perform 105 mL liquid swallows of an electrolyte solution at 30 s intervals after a 5 min adaptation in the supine position. The high-resolution array of pressure sensors and impedance channels dispersed across the entire anatomic region provided concurrent manometric and bolus transit dynamics. The HRIM results were detected using objective metrics, including integrated relaxation pressure (IRP), peristalsis analyzed based on absence, distal latency (DL), distal contractile integral (DCI), fragmentation, peristaltic break size (PB), contraction front velocity (CFV), and effective swallows. In addition, esophageal pressure topography was adopted for the presentation of HRIM data, which assigned colors to specific pressure levels that were then presented in a spatiotemporal plot [15]. The data were analyzed using package analysis software and interpreted by a practical thoracic surgeon dovetailed with Chicago Classification v3.0 [4].

### 2.5. Surgical Intervention

Treatment of achalasia is aimed at lowering the resting pressure of the LES [16]. Surgical myotomy of the muscle layer of the distal esophagus and LES, also known as Heller myotomy, is performed by 8 cm parallel cutting of the cardia muscle fibers on the anterior aspect above the LES and extending down to a small portion of the stomach, along with fundoplication to reduce the risk of postoperative gastroesophageal reflux [16,17,18]. The surgery was initially performed using an open procedure—either thoracotomy or laparotomy—and was then revolutionized into minimally invasive laparoscopic and robotic techniques. Currently, peroral endoscopic myotomy (POEM)—a form of natural orifice transluminal endoscopic surgery completed by creating a submucosal tunnel in the lower part of the esophagus to reach the inner circular muscle bundles of the LES in order to perform myotomy—is regarded as the endoscopic equivalent of Heller myotomy [19].

In our hospital, surgical myotomies were performed using either an open procedure or laparoscopic or robotic technology, with cutting of the anterior aspect of the distal esophagus and LES ranging from 4 cm above the EGJ to 2 cm below the EGJ. In addition, POEM was also performed in a small number of patients.

### 2.6. Statistical Analysis

Continuous data with normal distributions are expressed as the mean ± standard deviation (SD), and were compared using Student’s *t*-test. A relative frequency distribution consists of the relative frequencies or proportions of observations belonging to each category, expressed as percentages (%). All statistical analyses were performed using SPSS 18.0 (SPSS, Inc., Chicago, IL, USA).

## 3. Results

### 3.1. Patient Demographics and Exclusion Condition

From December 2014 to December 2019, 298 patients underwent HRIM at the TSGH due to symptoms of dysphagia. Twenty-one patients were excluded, including those with oropharyngeal dysphagia or secondary dysphagia, patients receiving prior upper gastrointestinal surgical interventions, and four patients younger than 20 years old. Moreover, one patient had pharyngeal dysphagia, two patients were diagnosed with corrosive injury of the esophagus with stricture, three patients were diagnosed with hiatal hernia with an anatomical abnormality at the cardia, one patient was diagnosed with anterior osteophytes of the cervical spine, one patient had chronic inflammatory demyelinating polyradiculoneuropathy, one patient was diagnosed with Zenker’s diverticulum, two patients with epiphrenic esophageal diverticulum received diverticulectomy, one patient with myasthenia gravis received video-assisted thoracoscopic thymectomy, two patients with esophageal malignancy received gastric tube reconstruction, and three patients had previous achalasia—two of whom received laparoscopic Heller myotomy, and the other of whom received robotic myotomy and Belsey Mark IV fundoplication. These exclusions led to a total of 277 patients (142 men; mean age, 54.7 years; median age, 56 years; range, 21–92 years) included in the final analysis.

### 3.2. Prevalence of Respective Esophageal Motility Disorders Based on the Chicago Classification v3.0

All study patients were categorized into eight groups based on the Chicago Classification v3.0 (Figure 1). Ineffective esophageal motility (*n* = 152, 24.5%) was the most common diagnosis by HRIM, followed by absent contractility (*n* = 38, 13.7%), EGJ outflow obstruction (EGJOO, *n* = 24, 8.7%), type II achalasia (*n* = 22, 7.9%), type I achalasia (*n* = 4, 1.4%), distal esophageal spasm (*n* = 1, 0.4%), and jackhammer esophagus (*n* = 1, 0.4%). Thirty-five patients (12.6%) had normal HRIM outcomes, indicating normal esophageal function. In addition, 26 patients (9.3%) were identified as having achalasia, and were further classified into three subtype groups as defined by the Chicago Classification v3.0 scheme, including 4 patients (1.4%) with type I, 22 patients (7.9%) with type II, and 0 patients with type III achalasia in our analysis.

### 3.3. Demographics, Clinical Characteristics of the Study Subjects, and Surgical Intervention Rates of Patients with Non-Spastic Motility Disorders

For non-spastic motility disorders, including type I and type II achalasia and EGJOO, surgical intervention with myotomy is frequently viewed as a first-line therapy. There were a total of 50 patients (52 ± 17.7 years) with non-spastic motility disorders in our study (Table 1), including 4 patients with type I achalasia, 22 patients with type II achalasia, and 24 patients with EGJOO. These patients were predominantly male (58%), and generally had a medium-to-thin body stature, with mean body height and body weight measuring 165.2 ± 9.7 cm and 64.5 ± 15.1 kg, respectively. Regarding the Eckardt score, dysphagia (1.74 ± 1.21 points) was the most frequently occurring symptom, followed by regurgitation (1.32 ± 1.02 points), weight loss (1.14 ± 1.089 points), and retrosternal pain (1.08 ± 0.99 points), contributing to an average stage II symptomatic demonstration (a total score of 5.28 ± 2.65 points). A total of 60% of patients underwent esophagoscopy, and 56% underwent upper gastrointestinal series. A total of 16 patients (32%) had undergone surgical interventions, including 2 patients (4%) with exploratory Heller myotomy, 8 patients (16%) with laparoscopic myotomy, 1 (2%) with robotic techniques, and 5 patients (10%) with peroral endoscopic myotomy. Based on the Chicago Classification v3.0, the criteria of non-spastic motility disorders include an elevated median IRP (>15 mmHg). Our study met the criteria, with IRP (mean ± SD) measuring 19.57 ± 9.39.

### 3.4. Demographics, Clinical Characteristics, and Surgical Outcomes of Patients with Non-Spastic Motility Disorders Who Received Surgical Interventions

Sixteen patients (32%) who underwent surgical interventions were further analyzed (Table 2), with a relatively young age (48.9 ± 16.4 years) and female predominance (male gender 43.8%) compared with the total non-spastic motility disorder group. These patients also had a relatively higher preoperative Eckardt score (a total of 5.44 ± 2.85 points), with frequently occurring dysphagia (1.94 ± 1.12 points), followed by regurgitation (1.44 ± 1.09 points), weight loss (1.13 ± 1.09 points), and retrosternal pain (0.94 ± 0.93 points). A total of 56.3% of patients who underwent surgery underwent esophagoscopy, and 87.5% underwent upper gastrointestinal series. Of all the surgical interventions, laparoscopic myotomy was performed in a maximum of eight patients (50%), followed by POEM in five patients (31.25%), exploratory techniques in two patients (12.5%) and robotic techniques in one patient (6.25%). Significant progress was noted after surgery, with a marked decrease in the postoperative 3-month Eckardt score (a total of 2.19 ± 1.11 points), including dysphagia (0.69 ± 0.48 points), regurgitation (0.56 ± 0.63 points), weight loss (0.44 ± 0.51 points), and retrosternal pain (0.50 ± 0.63), indicating obvious clinical success (delta: 3.25 ± 2.02 points). We also calculated the operative times (204.6 ± 56.6 min), related complication—including subcutaneous emphysema in one patient (6.25%) and aspiration pneumonia in two patients (12.5%)—postoperative course (7.6 ± 2.1 days), time to oral feed (2.0 ± 0.8 days), and time to hospital discharge (12 ± 5.7 days).

## 4. Discussion

Esophageal motility disorders comprise a heterogeneous group of diagnoses attributed to an imbalance of inhibitory and excitatory signaling of peristalsis in the esophageal body, along with impaired relaxation of the lower esophageal sphincter [20]. Decreased delivery of postganglionic transmitters—including nitric oxide and vasoactive intestinal peptide—at the level of the esophagogastric junction causes the failure of deglutition-induced LES relaxation in esophageal motility disorders [21]. The inception of HRIM, based on parameters of peristaltic vitality and timing as well as the function of the LES, contributed to further evaluation of esophageal symptoms and to subsequent refinement.

In our retrospective study, more than 80% of nonobstructive dysphagia patients were found to have esophageal motility disorders, with ineffective esophageal motility (IEM) being the most common diagnosis in our Taiwanese population. Our results are consistent with those of multiple studies, including a previous large cohort study by Alani et al. [22], which reported IEM to be the most prevalent esophageal motility disorder in academic community practice in the general population of the United States, and a study by Scheerens et al., who described IEM as the most frequently encountered esophageal motility disorder, found in 51% of 131 patients with symptoms of esophageal dysphagia referred for high-resolution manometry [23]. IEM, categorized as a minor disorder of peristalsis with the characteristic of impaired clearance based on the Chicago Classification v3.0 [4], is a manometric pattern found particularly in failed smooth muscle contraction in the distal esophagus, which is defined as more than 50% ineffective swallows with a DCI of less than 450  mmHg·sec·cm. Based on these findings and present data, IEM should always be considered during initial diagnosis, even though the treatment has been challenging due to the lack of promising promotional agents with a definite effect on improving bolus transit and restoring esophageal peristalsis [24]. However, a few studies have clarified achalasia as the most common diagnosis of nonobstructive dysphagia in the Taiwanese population [11]. The different contributive results between our study and the other study in the Taiwanese population might be due to the single-center study design, small sample size, and referral bias depending on the hospital reputation.

Regarding medical treatment for esophageal motility disorders, nitrates and calcium channel blockers are the most frequently used pharmacological drugs due to their characteristics as smooth muscle relaxants, which transiently reduce LES pressure. Nitrates increase the NO concentration in smooth muscle cells, which subsequently increases cyclic guanosine monophosphate levels, leads to dephosphorylation of the myosin light chain, and results in muscle relaxation [25]. Nifedipine inhibits LES muscle contraction by blocking cellular calcium uptake, and lowers the LES resting pressure by 30–60% [8,26]. However, the substantial drawback is that the improvement in dysphagia is usually incomplete, and these drugs are short-lived, with efficacy decreasing over time and commonly noticed side effects, including headache, dizziness, and pedal edema [25]. In contrast with the dubious effectiveness and the disadvantages of pharmacological medications, surgery is considered the most efficacious treatment for patients with non-spastic achalasia and EGJOO, due to its effective symptomatic palliation and prevention of disease progression. Dilation of the esophagus is commonly seen in non-spastic esophageal motility disorders, and approximately 5% of untreated achalasia cases will progress to end-stage disease with the development of a massively dilated and tortuous megaesophagus—also known as sigmoid esophagus—with irreversible elongation, dilatation, and loss of esophageal function [27]. Further deterioration of function can then lead to significant morbidity, including malnutrition, pulmonary complications from repeated aspiration, and chronic severe esophagitis [28]. In addition, compared to the general population, patients with achalasia have a 50 times greater risk of presenting esophageal squamous-cell carcinoma, which manifests 20–25 years after achalasia symptom onset [29]. Multiple mechanisms are related to the development of esophageal squamous-cell carcinoma in achalasia, mainly due to the stasis of gastric contents inducing squamous hyperplasia with papillomatosis and basal-cell hyperplasia, bacterial overgrowth, genetic alterations, and chronic inflammation [27,29]. Therefore, surgical intervention targeted at reducing the pathologically elevated pressure at the LES further relieves esophageal outflow obstruction [6], subsequently contributing to symptomatic relief and reducing the development of end-stage disease and malignancy. The timing of surgical intervention was not set by a specific cutoff point in current studies, and remained an individual and comprehensive decision. As for our study, for those patients who reported no symptomatic improvement after 12 months of medication treatment, we suggested surgical intervention.

Surgical myotomy of the muscle layer of the distal esophagus and LES—also known as Heller myotomy—has been a time-honored treatment for achalasia since it was first described in 1913 by the German surgeon Ernst Heller [18], with 8 cm parallel myotomies over the anterior and posterior aspects of the esophagus, dramatically improving quality of life. The two most important modifications to the original procedure include cutting of the cardia muscle fibers only on the anterior side, and the addition of fundoplication to reduce the prevalent risk of postoperative gastroesophageal reflux [16,17]. Myotomy is a lengthwise cut starting above the LES and extending down to a small portion of the stomach, with cuts through the external muscle layers while leaving the inner mucosal layer intact. The surgery was initially performed using an open procedure—either thoracotomy or laparotomy—and was subsequently revolutionized to minimally invasive laparoscopic surgery—which minimized risks and accelerated recovery significantly—and then to robotic technology, because of the advantages afforded by three-dimensional visualization. Currently, peroral endoscopic myotomy (POEM)—a form of natural orifice transluminal endoscopic surgery completed by creating a submucosal tunnel in the lower part of the esophagus to reach the inner circular muscle bundles of the LES in order to perform myotomy while preserving the outer longitudinal muscle bundles—is regarded as the endoscopic equivalent of Heller myotomy, with a minimal hospital stay [19].

Thirty-two percent of patients with non-spastic motility disorder (including type I and II achalasia and EGJOO) in our study had undergone myotomy under either laparotomy or laparoscopic, robotic, or peroral endoscopic techniques, which demonstrated a relatively lower rate of surgical intervention than that in a previous study by Shea et al., which reported 113 patients who underwent endoscopic surgical procedures among a total of 195 patients (57.9%) who had a diagnosis of EGJOO by HRIM [30]. This difference might be related to the relatively low symptomatic severity of the patients in our study, as well as the preference for noninvasive treatment in Eastern society. Additionally, our results reveal significant progress after surgery, with a marked decrease in the postoperative Eckardt score, indicating satisfactory clinical success, which is compatible with the conclusion demonstrated by Paula et al. that Heller myotomy allows an 89.6% decrease in symptoms and an increase in the quality of life of achalasia patients, based on a disease-specific quality-of-life questionnaire and the Eckardt score (*p* < 0.001) [31]. Similarly, Shea et al. showed that symptom resolution or improvement occurred in 94% of patients with EGJOO who underwent surgical myotomy [30]. Moreover, compared with the pre- and postoperative Eckardt scores, the dysphagia score had the largest decrease in our study, which was consistent with the findings of Oelschlager et al., who demonstrated a 95% improvement in dysphagia after standard myotomy in the long term (defined as 5 years and beyond) compared with slightly less efficacy (approximately 70%) for chest pain associated with achalasia. Based on these findings, we believe that the number of patients in our study who met surgical indications might have been underestimated, and the adequate timing for invasive intervention may have been postponed.

Of all the surgical interventions performed in our study, laparoscopic myotomy involved a maximum of 50% of patients, followed by 31.25% for POEM, 12.5% for exploratory techniques, and 6.25% for robotic techniques. A systematic review and meta-analysis by Campos et al. reported that laparoscopic myotomy combined with an anti-reflux procedure provided better symptom relief (90%) than other surgical approaches, as well as a low complication rate (6.3%) [6]. POEM, as a minimally invasive intervention, is considered to have comparable efficacy with that of laparoscopic myotomy, but a lower risk of perforation, bleeding, and infection [19]. Hungness et al. reported that POEM provided durable symptomatic relief in 94% of patients with non-spastic achalasia, with a low rate of complications at an average of 2.4 years of follow-up [32]. Only 5 patients underwent POEM in our study, with a relatively low percentage of 31.25% in a total of 16 patients who had undergone surgical intervention, which was associated with the recently developed (since November 2021) and unproficient techniques and the skill demand at our hospital. The results of the comparison between patients who underwent POEM and those who received other kinds of surgical myotomy at our hospital (Table 3) revealed a more apparent difference between pre- and post-POEM, with a greater delta. This indicates similar satisfactory clinical success of the transluminal endoscopic procedure and traditional surgery, even though the *p*-value discloses no significant difference. Moreover, there were no significant differences in operative time, complications, postoperative course, time to oral feed, or time to hospital discharge between patients who underwent POEM and those who received other kinds of surgical myotomy.

In a meta-analysis of over 7000 patients, including over 70 cohort studies, POEM was more effective than laparoscopic Heller myotomy in improving dysphagia at 12 and 24 months, although POEM patients were found to have a higher incidence of developing gastroesophageal reflux disease (GERD) symptoms [33]. Post-POEM gastroesophageal reflux is suggested to have a close to 50% incidence [34], which is significantly higher than that after laparoscopic Heller myotomy with fundoplication [35]. An endoscopic procedure such as transoral incisionless fundoplication (TIF) is an attractive option that creates an anti-reflux barrier through the creation of a valve 2–4 cm in length, with a 270 degree or greater circumferential wrap [36,37]. The other endoscopic option—the Stretta system—uses radiofrequency ablation to create a thermal effect below the mucosa at the EG junction and restore the reflux barrier [38]. Both adjunct endoscopic therapies offer less invasive options to replace surgical fundoplication, have fewer adverse effects, and do not limit future treatment options [39]. Therefore, further efforts to advance POEM should be mandatory in our hospital to yield successful outcomes in patients with esophageal motility disorders.

The present study has several limitations. First, this was a retrospective single-institution study, and the results may not be generalizable to represent the authentic prevalence of the disease, due to referral bias. Second, a possible interpretation bias was caused by the single-machine and single-physician interpretation of the objective results based on the clinician’s own opinion or intuition. Third, we did not assign a comprehensive standard protocol for detailed diagnosis, since only a small portion of the enrolled patients underwent esophagoscopy or upper gastrointestinal series, which might have revealed a defective condition during analysis. Fourth, there might have been measurement bias during the procedure, such as uneasy introduction of the catheter passing straightforwardly through the EGJ. Moreover, our study lacked a control group, with no follow-up Eckardt score for those who did not undergo surgery, so we could not carry out comparison between the surgical and non-surgical groups, and could only describe significant progress postoperatively. Furthermore, the sample size in the present study was relatively small, which may affect the feasibility of subgroup analyses and comparative statistical analyses. Finally, future studies with long-term follow-up are warranted, either analyzing surgical and non-surgical groups, evaluating subjects undergoing laparoscopic myotomy versus POEM, or comparing POEM with and without anti-reflux procedures, in order to determine the best therapeutic options for patients.

## 5. Conclusions

IEM was the most common diagnosis of nonobstructive dysphagia in our Taiwanese population based on HRIM and Chicago Classification v3.0, and should always be considered during initial diagnosis.

In addition, surgery is considered a safe and effective treatment for patients with non-spastic achalasia and EGJOO, due to its effective symptomatic palliation and prevention of disease progression; surgery also contributes to an obvious improvement of dysphagia compared with slightly less efficacy for other related symptoms. We believe that the number of patients in our study who met surgical indications might have been underestimated and that, consequently, the adequate timepoint for invasive intervention was postponed. Moreover, POEM, as a minimally invasive intervention, is considered to have comparable efficacy with that of laparoscopic myotomy, and warrants long-term follow-up studies either analyzing surgical and non-surgical group, evaluating discrepancy between laparoscopic and endoscopic interventions, or comparing POEM with and without anti-reflux procedures, in order to determine the best therapeutic options for patients.

## Figures and Tables

**Figure 1 jpm-12-00590-f001:**
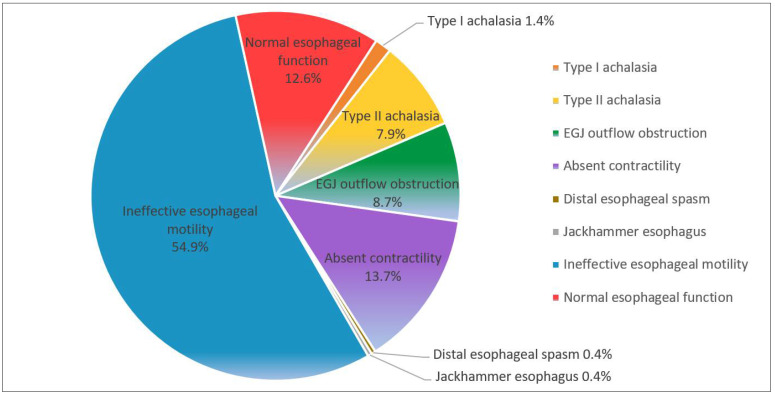
Prevalence of respective esophageal motility disorders based on Chicago Classification v3.0 scheme.

**Table 1 jpm-12-00590-t001:** Demographics, clinical characteristics of the study subjects, and surgical intervention rates of patients with non-spastic motility disorders.

Number of patients	N = 50
Age (mean ± SD)	52 ± 17.7
Male gender (%)	29 (58%)
Body height (mean ± SD)	165.2 ± 9.7
Body weight (mean ± SD)	64.5 ± 15.1
Smoking (%)	20 (40%)
Alcohol drinking (%)	16 (32%)
Eckardt score:	
Weight loss (mean ± SD)	1.14 ± 1.089
Dysphagia (mean ± SD)	1.74 ± 1.21
Retrosternal pain (mean ± SD)	1.08 ± 0.99
Regurgitation (mean ± SD)	1.32 ± 1.02
Total Eckardt score (mean ± SD)	5.28 ± 2.65
Objective metrics of HERM:	
IRP (mean ± SD)	19.57 ± 9.39 (−3, 43)
DCI (mean ± SD)	837.9 ± 1135.0
DL (mean ± SD)	5.58 ± 2.49
PB (mean ± SD)	4.02 ± 3.61
CFV (mean ± SD)	52.1 ± 114.9
Esophagoscopy (%)	30 (60%)
Upper gastrointestinal series (%)	28 (56%)
Surgical intervention:	
Exploratory Heller myotomy (%)	2 (4%)
Laparoscopic myotomy (%)	8 (16%)
Robotic myotomy (%)	1 (2%)
Peroral endoscopic myotomy (POEM, %)	5 (10%)

**Table 2 jpm-12-00590-t002:** Demographics and clinical characteristics of patients with non-spastic motility disorders who received surgical interventions.

Number of patients	N = 16
Age (mean ± SD)	48.9 ± 16.4
Male gender (%)	7 (43.8%)
Body height, cm (mean ± SD)	162.9 ± 6.5
Body weight, kg (mean ± SD)	62.8 ± 15.1
Smoking (%)	3 (18.8%)
Alcohol drinking (%)	3 (18.8%)
Preoperative Eckardt score:	
Weight loss (mean ± SD)	1.13 ± 1.09
Dysphagia (mean ± SD)	1.94 ± 1.12
Retrosternal pain (mean ± SD)	0.94 ± 0.93
Regurgitation (mean ± SD)	1.44 ± 1.09
Preoperative total Eckardt score (mean ± SD)	5.44 ± 2.85
Objective metrics of HERM:	
IRP (mean ± SD)	19.56 ± 12.17 (7, 38)
DCI (mean ± SD)	595.1 ± 697.6
DL (mean ± SD)	4.46 ± 2.04
PB (mean ± SD)	5 ± 3.97
CFV (mean ± SD)	70.3 ± 135.0
Esophagoscopy (%)	9 (56.3%)
Upper gastrointestinal series (%)	14 (87.5%)
Surgical intervention:	
Exploratory Heller myotomy (%)	2 (12.5%)
Laparoscopic myotomy (%)	8 (50%)
Robotic myotomy (%)	1 (6.25%)
Peroral endoscopic myotomy (POEM, %)	5 (31.25%)
Postoperative Eckardt score	
Weight loss (mean ± SD)	0.44 ± 0.51
Dysphagia (mean ± SD)	0.69 ± 0.48
Retrosternal pain (mean ± SD)	0.50 ± 0.63
Regurgitation (mean ± SD)	0.56 ± 0.63
Postoperative total Eckardt score (mean ± SD)	2.19 ± 1.11
Delta (mean ± SD)	3.25 ± 2.02
Operative time (mean ± SD)	204.6 ± 56.6
Complications	
Subcutaneous emphysema (%)	1 (6.25%)
Aspiration pneumonia (%)	2 (12.5%)
Postoperative course (mean ± SD)	7.6 ± 2.1
Time to oral feed (mean ± SD)	2.0 ± 0.8
Time to hospital discharge (mean ± SD)	12 ± 5.7

**Table 3 jpm-12-00590-t003:** Comparison of patients receiving POEM and other surgical myotomies.

	5 Patients Post-POEM	11 Patients after Other Kinds of Surgical Myotomy	*p*-Value
Preoperative Eckardt score	6.6 ± 1.34	4.9 ± 3.24	*p* = 0.162
Postoperative Eckardt score	2.20 ± 0.45	2.18 ± 1.33	*p* = 0.968
Delta Eckardt score	4.4 ± 1.14	2.73 ± 2.15	*p* = 0.063
Operative time	210.2 ± 76.8	202.0 ± 49.2	*p* = 0.799
Complications			
Subcutaneous emphysema	1 (20%)	0 (0)	
Aspiration pneumonia	0 (0)	2 (18%)	
Postoperative course	6.6 ± 0.5	8.1 ± 2.3	*p* = 0.189
Time to oral feed	2.2 ± 0.4	1.9 ± 0.9	*p* = 0.528
Time to hospital discharge	9.8 ± 3.4	13 ± 6.4	*p* = 0.317

## Data Availability

Data are contained within the article.

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
