# Peer review of "Impact of Surgical Intervention on Nonobstructive Dysphagia: A Retrospective Study Based on High-Resolution Impedance Manometry in a Taiwanese Population at a Single Institution"

_jpm, 2022, doi:10.3390/jpm12040590_

Round 1
Reviewer 1 Report
Dear Authors,
Congratulations on the very good work you are presenting.
I would like to comment two points:
- Why did you use the normal ranges from the CC3 whilst you used the unisensor catheter (InSIGHT Ultima® High-Resolution Manometry) which has different range of normal values including, the most important parameter in this study, the IRP. In this view, I would re-analyse the whole data and revise the findings all together.
Note from literature for unisensor catheter:
Normative values in the supine position: IRP highest 95th percentile ranged from 10.2 to 18.2 mm Hg, upright: the IRP 95th percentile in the upright position ranged from 10.8 to 15.8 mm Hg
in this study:
The data were analyzed using package analysis software and
143 interpreted by a practical thoracic surgeon dovetailed with Chicago Classification v3.0.
IRP (mean ± SD) 19.57 ± 9.39 (-3, 43) for the patient group.
Based on the Chicago Classification v3.0, the criteria of nonspastic motility disorders include an elevated median IRP (>15 mmHg). Our study met the criteria, with an IRP (mean ± SD) measuring 19.57 ± 9.39.
2nd comment
I could not see a comparison between the surgical group against those who did not undergo surgical intervention. How it was determined that those who did have intervention showed improvement?
Reviewer 2 Report
Dear Sirs
We read with attention the manuscript entitled “Impact of surgical intervention on nonobstructive dysphagia: A retrospective study based on high-resolution impedance manometry in a Taiwanese population at a single institution”
Although the authors should be congratulated for their work there are a few issues that should be addressed before considering this manuscript for publication.
First, there is a confusion between the title and the objective: while the title addresses the “…impact of surgical intervention…”, in line 85 of the Introduction the authors state that “The aim of our study was to elaborate the clinical characteristics of patients with nonobstructive dysphagia based on HRIM in a Taiwanese population and to analyze the therapeutic outcomes of those patients who ultimately underwent surgical interventions. Because of this proliferation of objectives, there is a lot of demographic and clinical characteristics of the patients in the results section that distract the reader from the study’s main objective. This data would be more appropriate for the Patients and Methods Section.
Second, the authors use a large retrospective cohort to investigate the role of surgery on non-obstructive dysphagia and conclude that “…surgery is considered the most effective treatment for patients with nonspastic achalasia and EGJOO due to its effective symptomatic palliation and prevention of disease progression.”
However, in the results section, the only information that the authors supply regarding the clinical outcome is that the Eckardt score was significantly lower after Heller myotomy, meaning that the operation is effective in controlling symptoms. In order to state that it is the most effective the results should be compared to a control group and there is no control group to compare to. Information regarding the clinical progress of the remaining 34 patients who were not operated on might help to substantiate the conclusions. Also, were there any operated patients in the spastic dysphagia group of patients? What was their outcome?
Additionally, information on the criteria used for selecting patients for surgical vs non-surgical treatment would be appreciated. What was the timing of surgical intervention? At what point of the clinical course it was decided to proceed with surgical treatment? Was there a minimum clinical Eckardt score? Or was the decision based on HRIM findings? HAw were the non-operated patients treated?
Regarding the operation itself: we agree that the number of operated patients is too small to allow for comparison among the different surgical approaches. However, information regarding operative time, complications, postoperative course, time to oral feed, and time to hospital discharge would also be appreciated in order to weigh the risk-benefit ratio of the surgical procedures. Finally, it is well known that Heller myotomy has a high recurrence rate in the long term. How long was the follow-up of the operated patients ( and of the eventual control group…). How long after the operation was the Eckardt score assessed?
In summary, it is clear the authors have gathered a fair amount of data on these patients. However, it is believed that this data should receive an extensive reorganization in order to substantiate the appropriate conclusions and to be accepted for publication in this journal
Round 2
Reviewer 2 Report
The authors have added some information regarding the patients' surgical and postoperative data and have made some small changes in the introduction and discussion sections. Unfortunately, the main issue of this manuscript (the lack of a control group) was not corrected by the authors. As a result, based on the presented data, the only scientifically acceptable conclusion would be that the Heller myotomy is safe and effective, on the short term, to treat non-spastic dysphagia. Any other conclusion would be beyond the boundaries of their results.
